# Potential Influence of Age and Diabetes Mellitus Type 1 on MSH2 (MutS homolog 2) Expression in a Rat Kidney Tissue

**DOI:** 10.3390/genes13061053

**Published:** 2022-06-13

**Authors:** Paško Babić, Natalija Filipović, Lejla Ferhatović Hamzić, Livia Puljak, Katarina Vukojević, Benjamin Benzon

**Affiliations:** 1Department of Medicine, University Hospital Dubrava, 10 000 Zagreb, Croatia; paskobabic@gmail.com; 2Department of Anatomy, Histology and Embryology, University of Split School of Medicine, 21 000 Split, Croatia; natalija.filipovic@mefst.hr (N.F.); katarina.vukojevic@mefst.hr (K.V.); 3Center for Applied Bioanthropology, Institute for Anthropological Research, 10 000 Zagreb, Croatia; ferhatoviclejla@gmail.com; 4Center for Evidence-Based Medicine, Catholic University of Croatia, 10 000 Zagreb, Croatia; livia.puljak@unicath.hr

**Keywords:** MSH2, diabetes mellitus, kidney

## Abstract

Background: Homeostasis of proliferating tissues is strongly dependent on intact DNA. Both neoplastic and non-neoplastic diseases have been associated with MSH2 (MutS homolog 2, a mismatch repair protein) deficiency. In this study, we examined how age and diabetes mellitus influence the expression of MSH2 in the kidney. Methods: To study the effect of age, three groups of healthy rats were formed: 2 months, 8 months, and 14 months old. Two groups of diabetic rats were formed: 8 months old and 14 months old. Expression of MSH2 in the kidney was studied by quantifying immunofluorescent staining. Results: Age was identified as the main factor that influences MSH2 expression in kidneys. The effect of age followed parabolic dynamics, with peak expression at 8 months of age and similar levels at 2 and 14 months. Diabetes had an age-dependent effect, which manifested as the increase of MSH2 expression in 14-month-old diabetic rats in comparison to healthy animals. Conclusions: Age influences MSH2 expression in the kidney more than diabetes mellitus. Since ageing is a risk factor for kidney neoplasia, downregulation of MSH2 in older rats might represent one of the pro-oncogenic mechanisms of ageing at a molecular level.

## 1. Introduction

Renal epithelium seems to be in a constant turnover throughout the lifetime [1,2]. This implies that, as in any proliferating cell, DNA repair systems have a pivotal role in the maintenance of cell homeostasis and orderly architecture of kidney tissue. One such system is involved in the repair of mismatched nucleotide bases, which normally occur during the replication process. The mismatch repair (MMR) system is composed of two groups of proteins, six MutS homologues (MSH1 to MSH6) and four MutL homologues (MLH1, PMS1, PMS2, and MLH3). As part of a complex mechanism, MutS homolog 2 (MSH2) and MutS homolog 6 (MSH6) heterodimers bind to single base-pair mismatches [3]. Successively, MutL homolog 1 (MLH1) and Post-meiotic segregation 2 (PMS2) heterodimers are recruited to this complex in order to complete the DNA repair process [3]. In addition to mismatch repair, MSH2 protein is upregulated by normal DNA replication and DNA damage caused by alkylating agents, ionizing radiation, and other oxidating agents [4,5,6,7,8]. Since tissue stem cell proliferation is usually driven by paracrine and endocrine factors, it has also been shown that the latter two upregulate MSH2 [9]. Furthermore, telomere shortening response and maintenance of chromosomal structural stability seem to be also mediated by MSH2 [10,11]. Given all of this, it might be hypothesized that MSH2 is involved in all of the processes that are thought to drive or mediate ageing [12].

Many studies indicate that diabetes, as well as other chronic inflammatory diseases, are pathological conditions characterized by oxidative stress damage as a result of increased levels of reactive oxygen species (ROS) and reactive nitrogen species (NOS), which exceed the anti-oxidative capacity of a cell [13]. The main source of oxidative stress reactors in diabetes is a chronically elevated glucose level which induces many diverse biochemical processes producing ROS and NOS [14]. These include glucose autoxidation, protein kinase C activation, methylglyoxal formation and consecutive glycation of proteins, increased hexosamine metabolism, sorbitol formation and oxidative phosphorylation [14]. Radical species, along with aberrant glycosylation cause structural, and thus functional, changes of proteins and DNA [15]. Structurally changed proteins can be either degraded by the ubiquitin system or refolded through the action of chaperon proteins [16]. On the other hand, oxidative damage triggers a complex protein network called cellular stress response, which includes upregulation of MSH2 protein expression, as well as upregulation of chaperons and other proteins that can prevent its degradation [6,8,17].

Major insights on MSH2 function in ageing and diabetes mellitus were derived from in vitro studies. Such studies have so far modeled very few combinations of factors influencing MSH2 expression. Furthermore, since ageing is a physiological process, it involves multiple factors that regulate MSH2 in amounts that are physiological and hard to simulate by means of *in vitro* studies [12]. Similarly, diabetes mellitus is a complex pathological process that can be only modeled *in vivo* in its full complexity [18]. Since both ageing, diabetes mellitus and MSH2 deficiency have been linked, to a certain extent, with neoplastic and non-neoplastic renal diseases [13,19,20,21,22], we decided to explore the influence of age and diabetes mellitus type 1, on renal MSH2 expression in an ageing diabetic rat model.

## 2. Materials and Methods

### 2.1. Animals and Diabetes Mellitus Model

Twenty-five male rats were raised under controlled conditions (22.1 °C temperature and 12/12 h light schedule) at the University of Split’s Animal Facility. Standard laboratory chow (4RF21 GLP, Mucedola srl, Settimo Milanese, Italy) and water were given *ad libitum*.

Diabetes mellitus type 1 (DM1) was induced in 2-month-old rats by intraperitoneal administration of streptozotocin (STZ; 55 mg/kg), freshly dissolved in citrate buffer (pH 4.5), after overnight fasting [23]. It usually takes a month for rats to develop first signs of complications caused by diabetes mellitus and to reach nonfluctuating levels of hyperglycemia above 300 mg/dl [23,24]. The age-matched control group received a pure citrate buffer solution. Five experimental groups were formed: 2-month-old control (*n* = 3), 8-month-old control (*n* = 7), 14-month-old control (*n* = 3), 8-month-old diabetic group (*n* = 6) and 14-month-old diabetic group (*n* = 6). Eight-month-old diabetic rats had DM1 for 6 months; 14-month-old diabetic rats had DM1 for 12 months. Plasma glucose levels were measured with a glucometer (One-133TouchVITa, LifeScan, High Wycombe, UK) once a month. DM1 rats with glycemia below 300 mg/dl were excluded from the study Diabetic rats received injections of 1 U of long-acting insulin (Lantus Solostar; Sanofi-Aventis Deutschland GmbH, Frankfurt, Germany) once a week, in order to prevent ketoacidosis.

The experimental protocol was approved by the Ethical Committee of the University of Split, School of Medicine. All experimental procedures followed the EU Directive (2010/63/EU).

### 2.2. Tissue Processing and Staining

Rats were anesthetized with isoflurane (Forane, Abbott Laboratories, Queenborough, UK) and perfused through the ascending aorta via the left ventricle with saline followed by Zamboni’s fixative [25]. The kidneys were removed, and tissue blocks were dehydrated upon fixation and embedded in paraffin wax [26]. After deparaffinization, tissue samples were run through the process of antigen retrieval in citrate buffer [26]. Nonspecific binding was blocked by Protein Block (Abcam, Cambridge, UK). Tissue sections were incubated with primary antibodies against MSH2 (Abcam, Cambridge, UK, diluted at 1:300) overnight at 4 °C. Staining was visualized by incubation with secondary antibodies labeled with green (donkey anti-mouse labeled with AF488, Invitrogen, Carlsbad, CA, USA, diluted at 1:400) fluorochrome. Finally, samples were counterstained with DAPI (4′,6-diamidino-2-phenylindole) [26].

### 2.3. Image Acquisition and Quantification

Photo-micrographs were shot by SPOT Insight digital camera (Diagnostic Instruments, Sterling Heights, MI, USA), mounted on Olympus BX61 fluorescence microscope (Olympus, Tokyo, Japan). Camera settings were set using image acquisition software CellA^®^ at 1360 × 1024 resolution, exposition of 1/333.3 s with a noise reduction filter. Ten micro-photographs of the kidney (5 in renal cortex and 5 in renal medulla), under the magnification of 200×, were shot per slide in green and blue fluorescent channels. Furthermore, fluorescence intensity histograms were acquired for the green fluorescence channel in ImageJ software (NIH, Bethesda, MD, USA) [27]. The region of the positive signal was determined by using the slides stained with secondary antibodies only, thus quantifying the autofluorescence and fluorescence due to the nonspecific binding of secondary antibodies. The region of the positive signal was defined as the one that excluded 99% of the signal obtained from fluorescence intensity histograms of slides stained with secondary antibodies only. Expression of MSH2 was quantified as the area under the curve (AUC) of fluorescence intensity histograms since this measure captures both total area under positive signal and fluorescent intensity of a signal [27]. For endogenous positive control, kidney tissue itself was used, since it is catalogued in the gene expression atlas as *MSH2* positive [28].

### 2.4. Statistical Analysis

Data are presented as an arithmetic mean and standard error of mean if not stated differently. Data were normalized to the average level of expression of the 2-month-old control group. Furthermore, differences between groups are expressed as fold change (i.e., the ratio of two means). Areas under curves (AUCs) and their interval estimates were calculated by using the AUC analysis routine in GraphPad Prism 8.0 software (Graph Pad, La Jolla, CA, USA). As a statistical measure of evidence, effect size and its 95% CI, R^2^ (η^2^), *p*-value and evidence (probability) ratios (E) between alternative (H_a_) and null (H_0_) hypotheses (i.e., models) based on differences in corrected Akaike information criterion (cAIC) were used [29]. Hypotheses were tested using a *t*-test, one-way and three-way ANOVA. In addition, *p*-values were interpreted according to ASA Statement on Statistical Significance and *p*-values [30]. All of the analyses were done in GraphPad Prism 8.0 software. Sample size was calculated by Mead’s resource equation.

## 3. Results

### 3.1. Dynamics of MSH2 Expression in Ageing Kidney of a Healthy Rat

Subcellular localization of florescent signal was in cytoplasm and in the nucleus in the form of discreet dots (Figure 1). The lowest expression of MSH-2 protein in the whole kidney of a healthy rat was found at 2 months of age. The peak expression of a 5-fold increase relative to the previous one was reached at 8 months. At 14 months of age, there was a drop in the expression of MSH-2 to levels similar to the initial ones (R^2^ = 91.35%, *p* < 0.0001, E ≈ 1.63 × 10^5^ in favor of H_a_) (Figure 1 and Figure 2a).

Since the cortex and medulla of the kidney have different physiological functions and are exposed to different microenvironments, with the medullary one being harsher, we decided to study MSH2 expression dynamics separately in those two tissue compartments [31,32]. The expression of MSH2 protein in the renal cortex showed the same pattern of expression as in the whole kidney. Likewise, the medullary expression of MSH2 had an almost identical profile (Figure 1 and Figure 2b,c). Renal medulla had 2.8-fold higher expression of MSH2 when compared to the cortex, in 2-month-old rats (95% CI 2 to 4.3-fold, R^2^ = 71.26%, *p* = 0.0345, E = 3.54 in favor of H_a_) (Figure 2d). On the other hand, MSH2 levels in the renal cortex of 8-month-old rats were 1.25-fold higher in comparison to the medullar levels (95% CI 1.01 to 1.49-fold, R^2^ = 30.86%, *p* = 0.0392, E = 2.53 in favor of H_a_). In 14-month-old rats, MSH2 seems to be expressed more in the cortex, but, unlike 8-month-old animals, the difference was much more uncertain (fold increase = 1.97, 95% CI 0.72 to 3.22, R^2^ = 53.86%, *p* = 0.0968, E = 14.58 in favor of H_0_) (Figure 2d).

### 3.2. Effects of Diabetes Mellitus on MSH2 Expression

Dynamics of MSH2 expression in the kidney of diabetic rats between the 8th and 14th month followed a descending trend, similar to the one in control rats. More specifically, MSH2 levels at 14 months were 1.63 times lower than levels of 8-month-old rats (95% CI 1.25 to 1.99-fold, R^2^ = 58.14%, *p* = 0.0038, E = 30.89 in favor of H_a_) (Figure 1 and Figure 2a). MSH2 expression in the cortex and medulla followed the same patterns as in the whole kidney (Figure 2b,c). When it comes to comparing medullary and cortical tissue compartments in diabetic rats that were 8 months old, MHS2 was 1.73 times more expressed in the cortex (95% CI 1.36 to 2.1, R^2^ = 66.61%, *p* = 0.0012, E = 115.3 in favor of H_a_). Likewise, MSH2 expression was 1.46 times higher in the cortex of 14-month-old diabetic rats than in medulla (95% CI 0.98 to 2-fold, R^2^ = 31.73%, *p* = 0.0565, E = 1.58 in favor of H_a_) (Figure 1 and Figure 2e).

When kidneys of diabetic rats were compared to healthy controls at each timepoint, then it appeared that there was no considerable influence on MSH2 expression at 8 months of age (fold decrease= 1.11, 95% CI 0.88 to 1.33, R^2^ = 9.86%, *p* = 0.296, E = 2.88 in favor of H_0_) (Figure 2a). On the other hand, at a 14-month timepoint, diabetes mellitus rats had increased expression levels of MSH2 in kidney by 2.01-fold (95% CI 1.15 to 2.87-fold, R^2^ = 52.79%, *p* = 0.0266, E = 2.659 in favor of H_a_) (Figure 2a). When the influence of diabetes mellitus was studied in the cortex, there seemed to be no clear differences in expression of MSH2 in both 8- and 14-month-old rats (Figure 2b). However, in the medullary compartment, diabetes decreased MSH2 expression by 1.38-fold (95% CI 1.16 to 1.6-fold, R^2^ = 56.86%, *p* = 0.0029, E = 41.75 in favor of H_a_) at the 8-month timepoint, whereas, in 14-month-old rats, diabetes increased expression of MSH2 by 2.29-fold (95% CI 1.46 to 3.12-fold, R^2^ = 66.08%, *p* = 0.0077, E = 11.77 in favor of H_a_) (Figure 2c).

Finally, in order to summarize the effects of the three factors (i.e., age, diabetes mellitus and tissue compartment) on the MSH2 expression, we used three-way ANOVA (Table 1). Age was found to explain 57% of differences in MSH2 expression, followed by tissue compartment, which could explain approximately 14% of the observed differences. Diabetes mellitus on its own was not associated with effects on MSH2 expression (Figure 2f); however, interaction of diabetes mellitus with age could explain approximately 7% of observed differences in data (Figure 2a–c).

## 4. Discussion

In this study, we have shown that MHS2 expression levels in kidneys depend on age and tissue compartment. Furthermore, we have shown that diabetes mellitus can somewhat modify those associations when it comes to both ageing and tissue compartment.

Many of the MSH2 expression dynamics in normal rats can be explained simply by hormonal changes in puberty and adulthood. Puberty in rats begins at 2 months of age and ends at 8 months [33]. During this period, a kidney’s weight increases by approximately 40% as a result of both hypertrophy and cellular proliferation [34]. Kidney growth is mediated by growth hormone and other endocrine and paracrine growth factors such as insulin-like growth factors (IGFs) [35]. Growth factors and cellular proliferation are both known to upregulate MSH2 expression [6,9]. Furthermore, as animals age more deeply into adulthood (14 month-old rats), levels of growth hormone decrease and oxidative damage progressively increases [34,36,37]; both factors might cause a reduction in an MSH2 expression level compared to that of a 14-month-old rat. Differences in MSH2 expression in tissue compartments are more puzzling. In adult rats (8 months and 14 months), MSH2 expression levels are higher in the renal cortex than in the medulla, which might be due to increased degradation of MSH2 caused by relatively hypoxic and consequently more oxidative medullary microenvironment [31,38]. On the other hand, we find it surprising that healthy pubertal rats had the opposite relationship between kidney compartments and MSH2 expression.

The effect of diabetes mellitus appears to depend on a timepoint and a tissue compartment. It seems that, in the cortex of 8-month-old rats, which were diabetic for 6 months, diabetes mellitus had no effect on MSH2 expression. The cortex itself in diabetic rats is known to initially undergo hypertrophy and hyperplasia, which is followed by extensive cellular and tissue damage after 6 months as a consequence of metabolic and hypoxic stress [39,40]. However, when it comes to MSH2 expression, our findings point to the possibility that up-regulatory stimuli such as cell proliferation due to hormonal factors, which are still intense at 8 months of age [33], and cellular stress response compensate for possible MSH2 degradation.

At 14 months of age and after 12 months of diabetes mellitus, MSH2 levels were higher in the cortex of diabetic rats. Since proliferative activity at that age is small [41], this might point to the hypothesis that the cellular stress response initiated by diabetes and its noxious consequences was successful in synthesizing MSH2 and preserving its levels to those that are higher than physiological ones. Expression differences in the medulla exhibited a similar but mitigated pattern.

Concerning expression of MSH2 in the renal medulla of 8-month-old diabetic rats, a decreased level of MSH2 might indicate two possible, mutually not exclusive, phenomena. A net effect of protein degradation despite the up-regulatory processes, the latter might be less efficient due to the demanding medullary microenvironment. On the other hand, since epithelial cells that comprise a loop of Henle express IGF-1 receptor and DM1 is known to downregulate IGF-1 and its receptor in kidney tissue, an effect of decreased IGF-1 paracrine and autocrine signaling and consequential decreased kidney growth might have occurred [35,42]. Theoretically, a telomere shortening can also upregulate MSH2 expression; however, kidney tissue in rats appears to age without telomere shortening [34].

### 4.1. Limitations of the Current Study

Given the aforementioned, it would be useful for future studies of MSH2 expression in ageing and diabetic kidneys to explore the correlation between MSH2 levels and oxidative stress markers (e.g., glutathione peroxidase, OGG1 and 8-oxo-dG), as well as the correlation between MSH2, paracrine growth factors (e.g., IGF-1 or its receptor), proliferation and cell damage markers. Furthermore, since gender has an influence on glucose homeostasis and diabetes mellitus (i.e., female sex seems to be a protective factor), future studies might also explore interaction of sex, diabetes mellitus and ageing on MSH2 expression in kidney tissue [43,44]. The results of current study would definitely benefit if they were corroborated with Western blot and RT-PCR quantification of MSH2 expression in kidneys.

### 4.2. Conclusions

When considering translational implications of our findings, so far, epidemiological studies have not characterized diabetes mellitus as an independent risk factor for kidney neoplasia [45,46]. This study yielded a result which is in agreement with epidemiological evidence, i.e., diabetes mellitus increased expression of MSH2 in older rats, which does not point to MSH2 dependent diabetes carcinogenesis.

In conclusion, it can be stated that, among studied factors, age is the primary factor that governs MSH2 expression in the kidney. Furthermore, a decrease in MSH2 expression in 14-month-old rats might be one of the pro-oncogenic molecular mechanisms of ageing, since age is the major risk factor for renal cancer [46].

## Figures and Tables

**Figure 1 genes-13-01053-f001:**
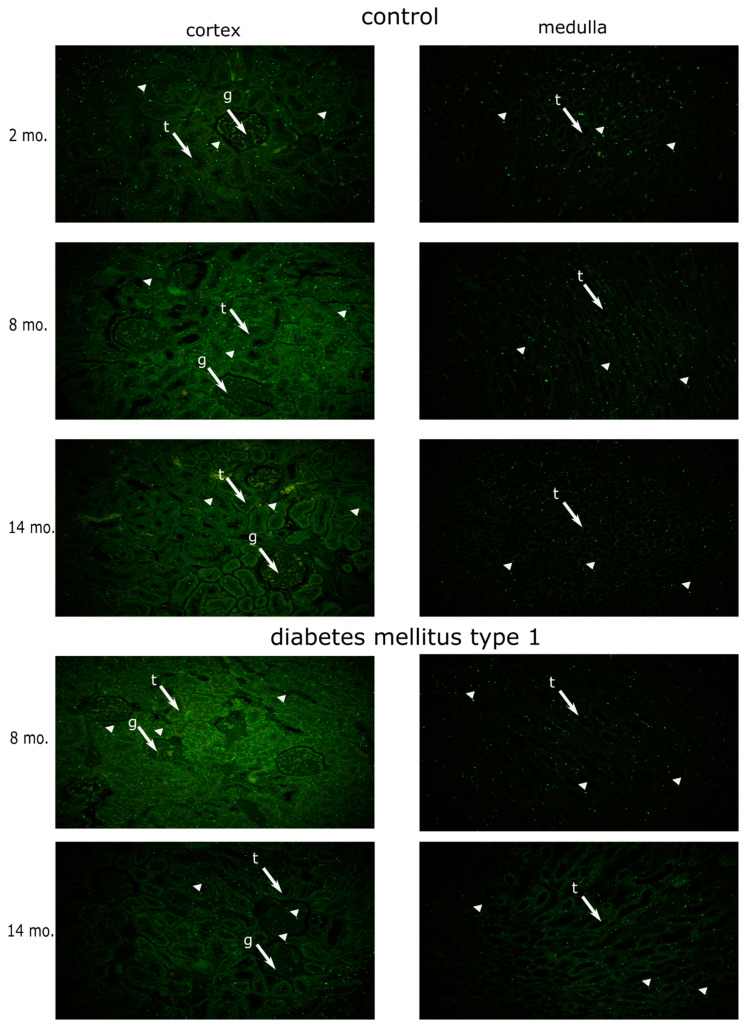
Representative microphotograph of MSH2 staining for each tissue compartment and timepoint in healthy and diabetic rats (200× magnification). Legend: g with arrow—glomerulus, t with arrow—tubulus, arrowhead—positive fluorescent signal for MSH2.

**Figure 2 genes-13-01053-f002:**
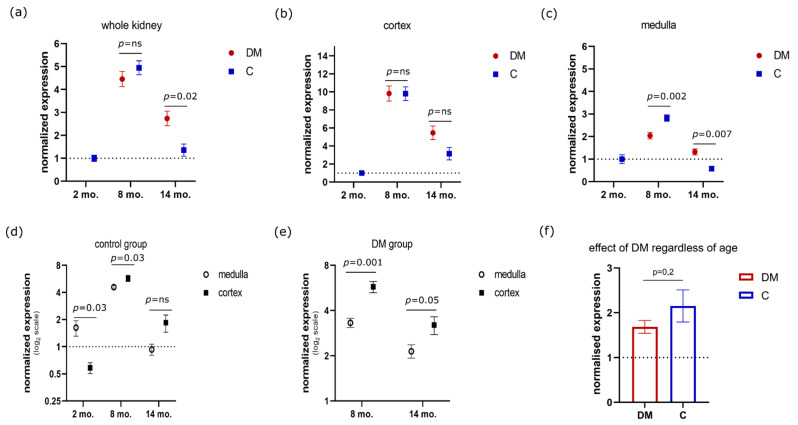
Normalized expression levels of MSH2 in whole kidney (**a**), cortex (**b**) and medulla (**c**) along with the direct comparison of expression levels between medulla and cortex in control (**d**) and diabetic groups (**e**). The last chart (**f**) represents a comparison of MHS2 expression in diabetic and healthy rats (C) regardless of age and tissue compartment. In each chart, expression was normalized to the level of a 2-month-old group, data are shown as the mean and standard error of mean. Rats in diabetes mellitus group (DM) were diabetic for 6 months in the case of the 8-month-old group and 12 months in the case of the 14-month-old group. All *p*-values presented in the figure were calculated by a *t*-test.

**Table 1 genes-13-01053-t001:** Effects of age, diabetes mellitus (DM) and tissue compartment on expression of MSH2.

*Factor*	*% of Total Variation* *Explained* *	*p-Value* *
**age**	**57.07**	**<0.0001**
**diabetes mellitus**	0.7612	0.2514
**tissue compartment**	**13.91**	**<0.0001**
** *Interactions* **		
**age × DM**	**6.581**	**0.0015**
**age × tissue compartment**	1.193	0.1531
**DM × tissue compartment**	0.9100	0.2107
**age × DM × tissue compartment**	0.5807	0.3154

* Three-way ANOVA with a fixed effect was used to calculate the effects of the factors.

## Data Availability

Raw data can be requested from the corresponding author.

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
