# Peer review of "Potential Influence of Age and Diabetes Mellitus Type 1 on MSH2 (MutS homolog 2) Expression in a Rat Kidney Tissue"

_genes, 2022, doi:10.3390/genes13061053_

Round 1

Reviewer 1 Report

1.       Gender biasness is known in diabetes. Authors should use both male and female rats in the study to rule out gender biasness. Please justify and discuss.

2.       Why diabetic group is not included for two months old rats? Please justify.

3.       How much time does it take to develop DM1 after streptozotocin injection? Please make it clear in the material and methods.

4.       Why do all the groups have a different “n” number of animals? Ideally, it should be the same. Please justify.

5.       Relevant references are missing in the first three sections of the material and methods. Please include.

6.       Figure2, Authors need to include the statistical details and significance in the figure legend.

7.       Gene expression studies have to include an endogenous positive control. What is the gene expression control in this study? Please justify and update the study accordingly.

8.       Authors just check the gene expression based on image quantitation. Please include a separate section, “Limitation of the current study”.

9.       Conclusions are missing. Please include.  

Author Response

Response to reviewer No. 1

Thank you for comments and suggestions, we think that they clarified certain points in the paper.

Below we will address all points in the order in which they were raised.

  • Gender biasness is known in diabetes. Authors should use both male and female rats in the study to rule out gender biasness. Please justify and discuss

Indeed, gender is an important factor that modifies glucose homeostasis mostly thorough influence of sex hormones on fatty tissue distribution and consequently on insulin resistance. Females seem to be protected from developing diabetes melitus. The principal aim of our study was to examine influence of age and diabetes mellitus on MSH2 expression in kidney [1][2]. In such set up we considered gender to be a confounding variable and, therefore we controlled for it by including only male rats in study. However, the downside of this approach is that we have not studied the effect of sex combined with diabetes mellitus on MSH2 expression and that poses a limitation of our study, so we included this issue in Limitations subsection of Discussion section.

  • Why diabetic group is not included for two months old rats? Please justify.

Rats that are younger than 2 months are considered to be prepubertal rats [3]. We did not want to study effects of diabetes melitus on rats that would model pediatric population, since overwhelming majority of diabetes melitus cases occur in adult age. Furthermore, streptozotocine (STZ) doses used to induce diabetes melitus in such young rats are 90 to 100 mg/kg which is higher than the doses used for adult or pubertal rats (55 mg/kg) [4]. Even though, administration of STZ to young rats almost immediately causes hyperglycemia, symptomatic complications develop 4 to 6 months later [4]. Thus, we concluded that introduction of 2-month-old diabetic rats to our study would introduce more confounding factors than the possible benefits in terms of conclusions and direct comparisons, therefore we omitted 2 mo. old diabetic group.

  • How much time does it take to develop DM1 after streptozotocin injection? Please make it clear in the material and methods.

Administration of 55 mg/kg of STZ to pubertal (young adult) or adult rats almost immediately causes hyperglycemia, which should reach a nonfluctuating level of 300 mg/dl after 28 days, on average. First chronic symptomatic complications are noticeable after one month. We clarified and referenced this in Methods section.

  • Why do all the groups have a different “n” number of animals? Ideally, it should be the same. Please justify.

Both diabetes melitus groups (i.e. 8 mo. and 14 mo. old) have 6 rats. When it comes to control groups we have 3 rats in 2 mo. old subgroup, 7 rats in 8 months old subgroup and again 3 rats in 14 mo. old subgroup. The reason for such asymmetric distribution of rats in control group is that we expected the differences in MSH2 expression should be the smallest between 8 mo. old diabetic and control group, so we increased numbers of mice in that 8-mo. control group in order to increase statistical power. This was expected because growth factors are known to mediate MSH2 expression and 8 mo. old rats are at the end of puberty, which itself is connected to high levels of growth factors in blood.

  • Relevant references are missing in the first three sections of the material and methods. Please include.

We included the references, as reviewer requested.

  • Figure2, Authors need to include the statistical details and significance in the figure legend.

We included statistical details in the legend of Figure 2 and on the figure itself, as review requested.

  • Gene expression studies have to include an endogenous positive control. What is the gene expression control in this study? Please justify and update the study accordingly.

Protein expression atlases are usually used to establish positive controls. According to European Bioinformatics Institute (EMBL-EBI) Gene expression atlas [5], MSH2 is expressed in kidney of rat at high and medium levels. Our stating agrees with the atlas. Furthermore, if subcellular localization is considered then the signal can be found in cytoplasm as well as in nucleus, which again is in accordance with Human proteome atlas, which contains data on subcellular localization. We have added a sentence about positive controls in both Methods section and Results section.

  • Authors just check the gene expression based on image quantitation. Please include a separate section, “Limitation of the current study”.

We agree that quantification by Western blot and RT-PCR would be beneficial to our study. The downside of these methods is that spatial resolution is lost and therefore we cannot track expression in kidney tissue compartments. We have added few sentences about this in “Limitations of current study” subsection in Discussion section.

  • Conclusions are missing. Please include.

We have added Conclusion subsection in the Discussion of our manuscript and abstract, as review requested.

References

[1]         M. Inoue, K. Inoue, and K. Akimoto, “Effects of age and sex in the diagnosis of type 2 diabetes using glycated haemoglobin in Japan: The Yuport medical checkup centre study,” PLoS One, vol. 7, no. 7, pp. 2–5, 2012, doi: 10.1371/journal.pone.0040375.

[2]         M. R. Meyer, D. J. Clegg, E. R. Prossnitz, and M. Barton, “Obesity, insulin resistance and diabetes: sex differences and role of oestrogen receptors,” Acta Physiol. (Oxf)., vol. 203, no. 1, pp. 259–269, Sep. 2011, doi: 10.1111/J.1748-1716.2010.02237.X.

[3]         P. Sengupta, “The laboratory rat: Relating its age with human’s,” Int. J. Prev. Med., vol. 4, no. 6, pp. 624–630, 2013.

[4]         M. A. Baig and S. S. Panchal, “Streptozotocin-Induced Diabetes Mellitus in Neonatal Rats: An Insight into its Applications to Induce Diabetic Complications,” Curr. Diabetes Rev., vol. 16, no. 1, pp. 26–39, Apr. 2019, doi: 10.2174/1573399815666190411115829.

[5]         M. Kapushesky et al., “Gene Expression Atlas update--a value-added database of microarray and sequencing-based functional genomics experiments,” Nucleic Acids Res., vol. 40, no. Database issue, Jan. 2012, doi: 10.1093/NAR/GKR913.

Reviewer 2 Report

Dear authors,

This study tries to analyze the influence of age and diabetes mellitus in MSH2 expression in cortex and medulla of rats’ kidney. The results are relevant and demonstrated that the age influences this expression more than diabetes mellitus and, furthermore, differences between MSH2 expression exist between medulla and cortex. The cited references are relevant to the research, the research design is appropriate, the methods are adequately described, the results are clearly presented, and the conclusions are supported by the results. However, minor changes are necessary from my point of view. Concretely, in the introduction, the authors indicate information related with its results (lines 63-64) which should not be presented in the introduction of the manuscript. In the next paragraph (lines 64-66), the authors should add a reference to support this information.

Author Response

Thank you for your fine comments of our work, we appreciate it very much.

We have deleted the lines 63-64 from introduction and added reference for line 64-66.

Round 2

Reviewer 1 Report

The authors successfully responded to the reviewer's comments/suggestions.